# Healthcare students' knowledge, attitude and perception of pharmacovigilance: A systematic review

**Monira Alwhaibi** [1,2]*, **Noha A. Al Aloola**[1,2]

**1** Department of Clinical Pharmacy, College of Pharmacy, King Saud University, Riyadh, Saudi Arabia,
**2** Medication Safety Research Chair, College of Pharmacy, King Saud University, Riyadh, Saudi Arabia

* malwhaibi@ksu.edu.sa

## Abstract

### Objective

The objective of this review is to evaluate the existing evidence about the knowledge, attitude, and perceptions (KAP) of healthcare students towards pharmacovigilance and adverse drug reactions reporting (ADRs).

### Methods

A systematic literature search was conducted using *MEDLINE*, CINAHL, EMBASE, ERIC, and Cochrane Database of Systematic Reviews via OVID. This review restricted the search to studies published in English from inception until December 2019.

### Primary and secondary outcome measures

The primary outcome was healthcare students' knowledge, attitude, and perceptions of pharmacovigilance.

### Results

Of the 664 articles identified, twenty-nine studies were included in the review. Overall, healthcare students vary in their knowledge and attitude towards pharmacovigilance and ADRs reporting. There was inconsistency in measuring KAP between the studies and the main drawback in the literature is lacking validated KAP measures.

### Conclusions

In summation, optimal KAP assessment can be achieved through developing a standard validated measure. Our future healthcare providers should have basics pharmacovigilance knowledge in order to rationally reporting ADRs and preventing serious health problems.

**Data Availability Statement:** All data generated or analysed during this study are included in this published article.

**Funding:** This research project was supported by a grant from the "Research Center of the Center for

Female Scientific and Medical Colleges", Deanship of Scientific Research, King Saud University.

**Competing interests:** The authors have declared that no competing interests exist.

## Introduction

Pharmacovigilance is an important discipline worldwide to ensure patient safety and the appropriate use of medicines [1]. World Health Organization (WHO) defines pharmacovigilance (PV) as "*the science and activities relating to the detection, assessment, understanding, and prevention of adverse effects or any other drug-related problem.*"[2] Pharmacovigilance and adverse drug reactions (ADRs) reporting education are important competencies all healthcare school students need to obtain before they graduate and be involved in clinical practice as healthcare professionals [3]. Therefore, educating healthcare students in the school of medicine, pharmacy, dentistry, or nursing and involving them early in clinical practice to prescribe, administer, and/or monitor medications is essential to ensure the safe use of medications [4].

Healthcare students may not recognize the importance of post-marketing ADR and may not have received sufficient knowledge and skills to recognize and adequately report the ADRs during their education. Literature indicated that many healthcare students missed the training on this topic, and inadequately prepared during their education for their role in monitoring and reporting ADRs [3, 5]. In addition, previous studies have shown that pharmacy students have insufficient knowledge of pharmacovigilance and ADRs reporting [3, 5].

Unfamiliarity with pharmacovigilance and ADR-reporting have been associated with ADRs underreporting by healthcare professionals [6, 7]. Further, underreporting and the lack of understanding of ADRs could lead to a greater burden on patients, payers, and healthcare systems. Therefore, knowledge and perception toward the safety profile of medications are essential. Educating healthcare professionals on the possible existence of unexpected adverse reactions and how to report them to the local regulatory authorities can facilitate the detection and assessment of drug safety signals.

The purpose of this systematic review is to evaluate the literature that measures the level of healthcare students' knowledge, attitude, and perception of pharmacovigilance and ADRs reporting. This can help to identify the current need for education/training on pharmacovigilance and the research need to improve our understanding of healthcare students' knowledge, attitude, and perception of pharmacovigilance.

By summarizing the published literature in this area it should be possible to grasp a more understanding of the existing evidence and understand future needs for research in this area. Our review research questions are:

1. What is known about healthcare students' pharmacovigilance knowledge?

2. What is known about healthcare students' attitudes and perceptions of pharmacovigilance?

3. Are there any validated measures to assess students' knowledge, attitude, and perception toward PV in the existing literature, and what could future studies add to our understanding of the healthcare students' knowledge, attitude and perception of pharmacovigilance?

## Methods

### Eligibility criteria

Study inclusion criteria were as follows: (1) study population consisting of healthcare students (medical, pharmacy, dental, and nursing) at any stage of their undergraduate training, (2) the study outcome is the knowledge, attitude or perception of pharmacovigilance, and (3) study design is cross-sectional.

Exclusion criteria were as follows (1) study population consisting of postgraduate or healthcare professionals, (2) qualitative study design, and (3) report language is non-English.

## Search strategy

The present systematic review was reported according to the Preferred Reporting Items for the Systematic Reviews and Meta-Analyses (PRISMA) guidelines (S2 Appendix) [8].

Research articles were retrieved from six databases (MEDLINE via EBSCO, Cumulative Index to Nursing and Allied Health Literature (CINAHL®) via EBSCO, EMBASE, ERIC via EBSCO, and Cochrane Database of Systematic Reviews via OVID) with database-specific queries. These databases were searched using both controlled and free-text language. In terms of free-text search, the keywords included the following terms/combination of terms: (knowledge OR attitude OR perception) AND (healthcare students OR medical students OR pharmacy students OR dental) AND (pharmacovigilance OR adverse drug reactions reporting). For the controlled language search included the following exploded Medical Subject Headings (MeSH) terms: "knowledge", "attitude", "perception", "Students, Medical", "Students, Pharmacy", "Students, Nursing", "Students, Dental", "Pharmacovigilance" and "Drug-Related Side Effects and Adverse Reactions" as recommended for each databases. S1 Appendix shows the complete search strategy used in MEDLINE.

Articles search of the four databases was conducted independently by two review authors (MA and NA), any disagreement was resolved by consensus. Besides, bibliographies from the selected articles were searched manually for relevant articles. Limits that were applied included selecting studies published in English from inception until December 2019; studies pertaining to the evaluation of pharmacovigilance knowledge, attitude or perception and where the participants are healthcare students.

## Study selection

At first, literature screening of the extracted articles involved examining the titles and abstracts for relevant articles for inclusion was conducted independently by two review authors (MA and NA). Then, the review authors evaluated the full-text articles against the inclusion/exclusion criteria. The article selection process resulted in twenty-nine studies included in this systematic review (Fig 1).

## Data extraction and quality assessment

The two review authors (MA and NA) independently extracted the included data. Information about the study characteristics, methodological details, main outcome measures, and findings were extracted from the selected articles and organized in an excel table to facilitate the assessment of their quality using STROBE (Strengthening The Reporting of OBservational Studies in Epidemiology) checklist [9]. This tool covers twenty-two criteria for study design quality and biases in the study. For each criterion met, the study gets one point; the highest score indicates the highest quality of the study. Besides, we have used the five items risk of bias in cross-sectional surveys of attitudes and practice [10].

## Statistical analysis

The published literature was analyzed qualitatively and the results (number and percentage) were reported in a narrative way, focusing on common findings that we identified across the included studies.

## Patient and public involvement

Patients and the public were not involved in the design or conduct of this study.

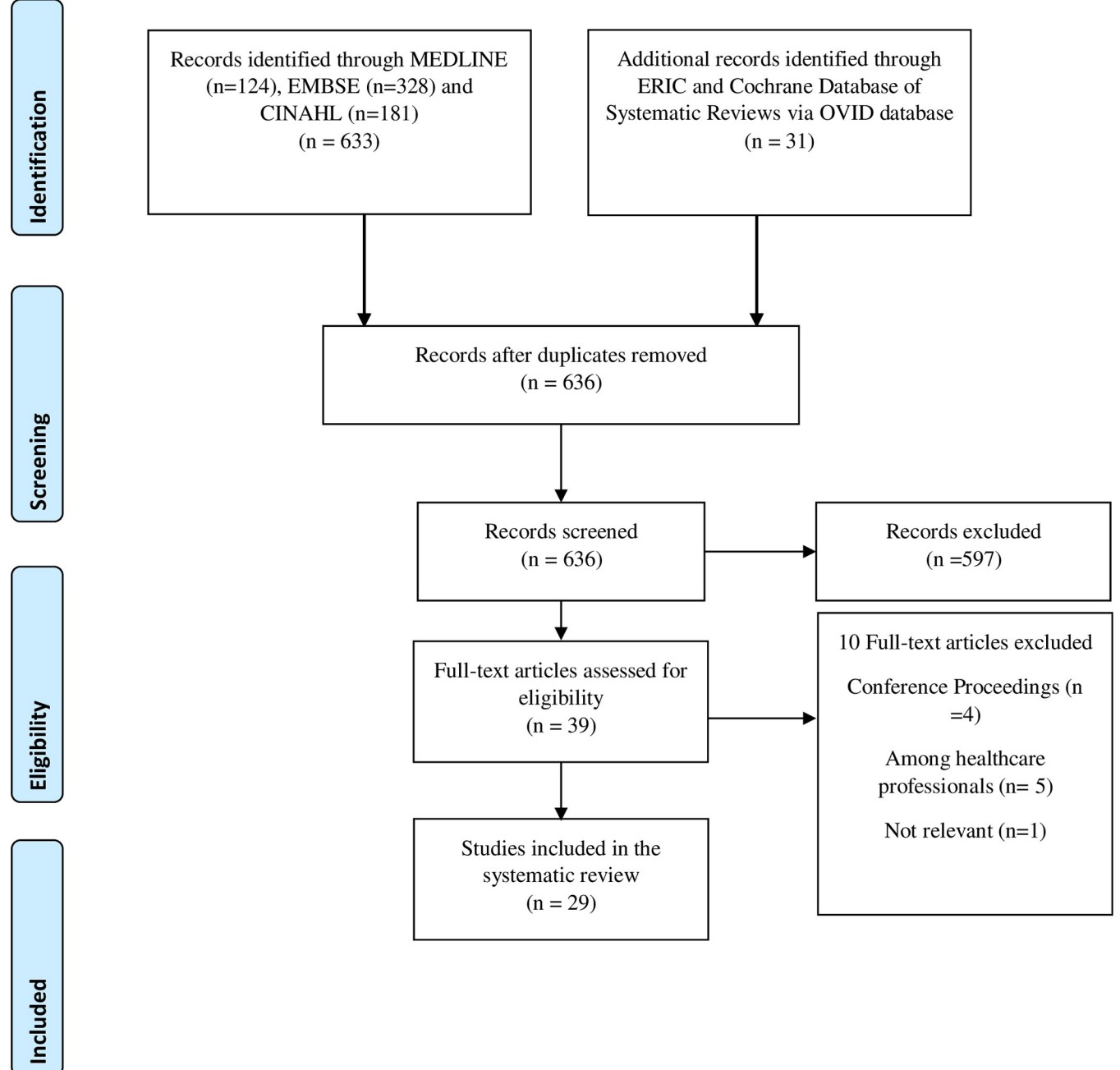

From: Moher D, Liberati A, Tetzlaff J, Altman DG, The PRISMA Group (2009). *P*referred *R*eporting *I*tems for *S*ystematic Reviews and *M*eta-*A*nalyses: The PRISMA Statement. PLoS Med 6(7): e1000097. doi:10.1371/journal.pmed1000097

**For more information, visit www.prisma-statement.org.**

**Fig 1. PRISMA flow diagram.** *From*: Moher D, Liberati A, Tetzlaff J, Altman DG, The PRISMA Group (2009). *P*referred *R*eporting *I*tems for *S*ystematic Reviews and *M*eta-*A*nalyses: The PRISMA Statement. PLoS Med 6(7): e1000097. doi: 10.1371/journal.pmed1000097 For more information, visit www.prisma-statement.org.

## Results

### Selection process

A total of 664 articles were identified from all searches, with 636 remaining after the removal of duplicates. Based on the title and abstract screening, 39 full-text articles were retrieved. Of these, twenty-two studies met the inclusion criteria and included in the final review (Fig 1).

### General characteristics of included studies

The characteristics of the included studies are displayed in Table 1. Studies that evaluated pharmacovigilance knowledge, attitude or perception of healthcare students started to appear since 2011. The majority of the included studies involved pharmacy students [11–19], while few studies specifically focused on healthcare students [20–22], nursing students [23], and dental students [24, 25]. The total number of included studies was twenty-nine, with sample sizes ranging from 30 to 874. Response rates across the studies varied from 24% to 100%. Twenty-seven studies examined the Pharmacovigilance knowledge [11–35], twelve studies evaluated attitudes towards pharmacovigilance [14, 16, 19–21, 23–26, 29, 30, 33], and fourteen studies measured the perception of pharmacovigilance [11–15, 17, 18, 21–23, 26–28, 31, 36]. Around 50% of the included studies (n = 15) were pilot tested among the students that were not included in the final analysis. A pilot test was used to evaluate the validity and reliability of the instrument with an overall reliability coefficient, Cronbach's alpha, ranging from 0.69–0.82 suggesting that the items included in these studies have relatively high internal consistency.

### Quality assessment

Overall, the included studies scored well for clearly stating the study aims, design, target population, risk factors and outcomes measurement, result explanation, and discussion and conclusion justified by the results (Table 1). The main issues were mainly related to failure to address the sample size calculation or addressing the non-response rate. Besides, many studies have no information about the missing data within completed questionnaires and have not conducted pilot testing.

### Main findings

Key information form selected articles were extracted and tabulated using the following categories: pharmacovigilance knowledge, attitude or perception (Table 1).

### Pharmacovigilance knowledge

Knowledge about pharmacovigilance and ADR reporting was mainly evaluated using multiple-choice response options ranging from 10 to 15 questions [11, 12, 14–17, 19, 20, 23, 26, 29–35]. A score of 1 was given for each correct answer and 0 for each wrong answer. The most common questions asked about PV, ADR definitions and the local regulatory body of ADR reporting, while few studies asked about the ADR causality assessment, types of ADR, and the online WHO PV database (S3 Appendix). Generally, knowledge of healthcare students about the local regulatory body and local reporting system of ADR reporting was inadequate. One study conducted by Khan et al compared the knowledge between medical and pharmacy students and found that pharmacy students have a significantly higher knowledge of pharmacovigilance compared to medical students [26]. Another study by Sivadasan et al had also compared medical to pharmacy students and found that pharmacy students have better knowledge and perception towards pharmacovigilance and ADR reporting compared to medical students [12].

**Table 1. Characteristics of included studies that measures knowledge, attitude, and perceptions of medical students.**

| Author (Publication Year) | Country | Study Design | Student Type | Total Students | Response Rate | Questionnaire Development | Outcomes | Main Findings | Quality * |
|---|---|---|---|---|---|---|---|---|---|
| Katyal et al (2019)[33] | India | Cross-sectional study | Medical students | 253 | | Was developed based on reviewing the literature | Knowledge and Attitude | Around 60% were familiar with the term 'Pharmacovigilance' | >75% |
| Marko (2019) [29] | India | Cross-sectional study | 2nd year, pre-final year, and intern Medical students | 228 | | Pretested questionnaire. | Knowledge and Attitude | > 60% of responders have knowledge about PV and ADRs. Around 80% have positive attitude toward ADR reporting. | <75% |
| Yu et al (2019)[32] | South Korea | Cross-sectional study | Pharmacy students | 303 | | The survey was developed based on a mixed theoretical model | Knowledge and Attitude | Around 67% have knowledge regarding PV, attitude towards ADRs range from 30% to 78% | >75% |
| Khan et al (2018)[34] | Pakistan | Cross-sectional study | Final year pharmacy students | 122 | | Well designed and structured questionnaire | Knowledge | Overall mean score of knowledge about ADRs and PV was 7.46 ± 2.19 | <75% |
| Gaude et al (2018)[30] | India | Cross-sectional study | Final year medical students | 95 | 73% | Predesigned questionnaire | Knowledge and Attitude | Around 55% of student answered the questions related to knowledge correctly and 67.3% had a positive attitude | >75% |
| Ajantha et al (2018)[31] | Chennai | Cross-sectional study | Dental students | 200 | | A validated questionnaire | Knowledge and Attitude | The knowledge regarding PV and ADRs was low ranging from 29% to 32% | <75% |
| Aamir et al (2018)[35] | Pakistan | Cross-sectional study | Pharmacy and Medical Students | 2010 | | Well-structured questionnaire | Knowledge | Poor knowledge towards PV and ADRs reporting was noticed among medical and pharmacy students | >75% |
| Tadvi et al (2018)[20] | Saudi Arabia | Cross-sectional study | 3rd and onwards medical students | 148 | 59% | Was developed based on the information obtained from previous studies | Knowledge and Attitude | The knowledge regarding PV and ADRs was low ranging from 28% to 57%. | >75% |
| Limaye et al (2018)[38] | India | Cross-sectional study | Pharmacy students | 352 | 88% | Was developed based on reviewing the literature and pilot tested among 30 students | Knowledge and perception | PV knowledge (44%) and perception (58%) | <75% |
| Chhabra et al (2017)[24] | India | Cross-sectional study | 3rd and final year dental students | 241 | 88% | Was developed from Theory, research, observation, and expert opinion and pilot tested among 35 students (Cronbach's alpha was 0.72 for knowledge; 0.86 for attitude) | Knowledge and Attitude | The total median PV knowledge score was 6, total median attitude score was 35 | >75% |
| Alkayyal et al (2017)[16] | Saudi Arabia | Cross-sectional study | 4th, 5th, and 6th year pharmacy students | 259 | | Was developed based on extensive literature search and pilot tested among 10 students | Knowledge and Attitude | The mean PV knowledge score was 4.15 | >75% |
| Othman et al (2017)[39] | Yemen | Cross-sectional study | Final year pharmacy students | 385 | 92% | Was developed based relevant literature and pilot tested among 20 students | Knowledge and Perception | The knowledge regarding PV and ADRs range from 20% to 81%, perceptions range from 60% to 97% | >75% |
| Al-Shekaili et al (2017) [17] | Oman | Cross-sectional study | Final year pharmacy students | 118 | 79% | Was developed based on reviewing the literature | Knowledge and Perception | The PV knowledge range from 17% to 78%, perceptions range from 39% to 80% | >75% |

*(Continued)*

**Table 1.** (Continued)

| Author (Publication Year) | Country | Study Design | Student Type | Total Students | Response Rate | Questionnaire Development | Outcomes | Main Findings | Quality * |
|---|---|---|---|---|---|---|---|---|---|
| Osemene et al (2017)[15] | Nigeria | Cross-sectional study | Final year pharmacy students | 342 | 98% | The study adapted the survey instruments used in similar studies, Pilot tested (Cronbach alpha = 0.72) | Knowledge and Perception | The mean PV knowledge score was 4.3, mean perception scores range from 1.8–4.6 | >75% |
| Schutte et al (2017)[21] | Netherlands | Cross-sectional study | 3$^{rd}$ to 6$^{th}$ year medical students | 874 | 7–24% | Was developed and pilot tested | Knowledge and Attitudes | Knowledge regarding PV and ADRs range from 28% to 95% | >75% |
| Rajiah et al (2016)[18] | Malaysia | Cross-sectional study | Final year pharmacy students | 108 | | Was designed after a detailed review of relevant literature was pilot-tested among 20 pharmacy students (Cronbach's alpha = 0.82) | Knowledge and Perception | Knowledge regarding PV and ADRs range from 7.4% to 92% and perception range from 3% to 75% | >75% |
| Abubakar et al (2015) [22] | Nigeria | Cross-sectional study | 4$^{th}$ and 5$^{th}$ year medical students | 108 | 74% | The questions were extracted from previous literature and pilot-tested among 20 medical students (Cronbach's alpha = 0.69) | Knowledge and Perception | Knowledge regarding PV and ADRs range from 10% to 94%, perception range from 6% to 98% | >75% |
| Farha et al (2015)[27] | Jordan | Cross-sectional study | 4$^{th}$, 5$^{th}$, and 6$^{th}$ year pharmacy students | 225 | 67% | A questionnaire previously developed by the study | Knowledge and Perception | The knowledge regarding PV and ADRs was ranging from 5% to 65% | >75% |
| Kothari et al (2015)[28] | Anand district in Gujarat, India | Cross-sectional study | 5$^{th}$, and 6$^{th}$ year pharmacy students | 300 | | Was developed based on the literature and pilot-tested among 25 pharmacy students | Knowledge and Perception | Knowledge regarding PV and ADRs range from 13% to 61% | <75% |
| Khan et al (2015)[26] | Pakistan | Cross-sectional study | Final-year pharmacy and medical students | 91 | | Was designed by the authors after an extensive literature review and pilot tested among 10 students (Cronbach's alpha = 0.81) | Knowledge, attitude and Perception | PV knowledge range from 31–91% for pharmacy and 7–84% for medical students, Attitude range from 47–98% for pharmacy and 35–98% for medical students | >75% |
| Shalini et al (2015)[25] | Malaysia | A pilot study | Pre-final and final year dental students | 61 | 76% | Was adapted from the previously published paper and pilot-tested among 20 students (Cronbach's α = .73) | Knowledge and Attitude | Knowledge regarding PV range from 3% to 50% | <75% |
| Jha et al (2014)[40] | Nepal | Cross-sectional study | Pharmacy students | | | Was developed after consulting previous studies | Knowledge | | >75% |
| Sivadasan et al (2014) [12] | Malaysia | Cross-sectional study | Pre-final and final year medicine and pharmacy students | 479 | 63% | Questionnaire was prepared from the literature and pilot-tested among 20 pharmacy students (Cronbach's alpha = 0.72) | Knowledge and Perception | Knowledge regarding PV range from 21% to 59% for pharmacy students and 6% to 72% for medical students, | >75% |

(*Continued*)

**Table 1.** (Continued)

| Author (Publication Year) | Country | Study Design | Student Type | Total Students | Response Rate | Questionnaire Development | Outcomes | Main Findings | Quality * |
|---|---|---|---|---|---|---|---|---|---|
| Reddy et al (2014)[14] | India | Cross-sectional study | Pharmacy students | 225 | 90% | Was generated and adapted from previous studies and pilot-tested among 15 students (Cronbach's alpha = 0.72) | Knowledge, Attitude and Perception | The knowledge regarding PV and ADRs was ranging from 5% to 65%, perceptions from 40% to 95% | <75% |
| Sivadasan et al (2014) [23] | Malaysia | A pilot study | Pre-final and final year Nursing Students | 32 | 100% | The questionnaire was adapted from the previously published paper and pilot-tested among 20 students (Cronbach's alpha = 0.73) | Attitude | The mean score on PV knowledge was found to be 12.31, for attitude was 15.1, for perception was 15.06. | >75% |
| Sharma et al (2012)[13] | Punjab, India | Cross-sectional study | Final year pharmacy students | 30 | | The questionnaire was adapted from the previously published study | Knowledge and Perception | The knowledge regarding PV and ADRs was ranging from 10% to 90%. | <75% |
| Gavaza et al. (2012)[19] | United States | Pilot study | Final year pharmacy students | 58 | 91% | A survey instrument adapted from previous research | Knowledge and Attitude | The PV knowledge score range from 29% to 82% and mean score on PV attitude was 5.2. | >75% |
| Elkalmi et al (2011)[11] | Malysia | Cross-sectional study | Final-year (fourth-year) pharmacy students | 510 | 84% | Was developed from the literature and a qualitative study a pilot-tested to a sample of 20 pharmacy students (Cronbach's alpha = 0.76). | Knowledge and Perception | The knowledge regarding PV and ADRs was ranging from 17% to 96%, perceptions from 40% to 95% | ~75% |
| Kalari et al. (2011)[36] | United States | Cross-sectional study | Second and third year pharmacy students | 228 | 65% | | Perception | The perception regarding PV and ADRs was ranging from 25% to 84% | <75% |

PV: Pharmacovigilance

*Quality of the studies was evaluated using the STROBE (Strengthening The Reporting of OBservational Studies in Epidemiology) checklist

## Pharmacovigilance attitude

Attitude towards pharmacovigilance and ADR reporting was measured using multiple-choice response options ranging from 2 to 5 questions [20, 30], and using a 5-point Likert scale [16, 24]. Although only a few studies evaluated attitude; it should be noted that some questions used were related to measuring the perception rather than attitude [16, 22, 26, 31]. The most commonly asked question was about the willingness of students to report any ADR in their future practice (S3 Appendix). The attitude towards PV and ADRs from the included studies ranged from 25% to 97% [16, 20, 24].

## Pharmacovigilance perception

Perception about pharmacovigilance and ADR reporting was measured using a 5-point Likert-scale format (1 strongly agree to 5 strongly disagree) [11–13, 15, 17, 21, 23], questions range from 5 to 13 questions [11–13, 23, 31]. However, some studies evaluated perceptions using multiple-choice response options [14]. The majority of included studies evaluated perceptions by asking students about their belief of PV, i.e. if they think that PV should be

included in the curriculum, if they think ADR reporting should be made compulsory, and if they are allowed or trained to perform ADR reporting during the clerkship (S3 Appendix). The perception regarding PV and ADRs from the included studies was ranging from 25% to 97% [11–13, 15, 17, 21, 23]. Khan et al compared the perceptions between medical and pharmacy students and found that pharmacy students reported more positive responses to all of the perceptions statements than the medical students (P<0.05) [26].

## Discussion

The present systematic review identified the available literature that evaluated KAP of PV and ADR reporting of any healthcare school students. Most of the published studies evaluated KAP among pharmacy students, while few focused on medical, dental, and nursing students. A summary of different measures used to assess KAP among different healthcare students was provided in this review.

Our review highlights the main drawback in this area, which is the lack of standardized validated measures to assess knowledge, perception, and attitude toward PV and ADR reporting. There were variations in items used in different studies to assess KPA of students. The survey instruments were pilot tested in fifteen studies and the internal consistency measured using Cronbach's Alpha. However; no item analysis in the form of difficulty and discrimination was reported in any of these studies.

Moreover, findings from this review highlight the variation in KAP of different healthcare students towards pharmacovigilance and ADRs reporting. This variation could be attributed to the following factors: i) inconsistency in tools/measured used to evaluate KAP between the studies; ii) lacking validated KAP measures, iii) different study setting, and iv) different PV and ADR reporting experiences during their education. Therefore developing a standard validated measure is needed to optimize KAP assessment.

Our knowledge findings are consistent with the previously published review, which investigates the pharmacovigilance competencies of all healthcare students [3]. Knowledge about PV is poor, despite the good perception about PV importance and the good attitude toward PV and ADR reporting. Based on *Reumerman et al.* review many factors could influence PV competencies such as; type of healthcare school, academic level of study and previous training [3]. This review shows that educational interventions such as; short lectures, workshops, training in ADR reporting and assessment have improved healthcare students' knowledge, perception and positive attitude toward PV [3]. However, it is unknown which education intervention was the best to improve students' PV knowledge and competences, due to variation in questions or outcomes scores that have been used by the authors.

In our review, students' satisfaction towards PV coverage in their curriculum varied from 21 to 85%. This finding indicates variation in PV integration in the curriculum between different healthcare schools. Besides, the finding highlights the need for uniform PV educational intervention. Given the importance of PV and ADR reporting in preventing serious health problems, more education in the field of PV and ADRs is needed. Moreover, standards for teaching PV have been developed by experts working in different fields of medication safety worldwide in the World Health Organization (WHO and the International Society of Pharmacovigilance (ISoP). The WHO-ISoP group had created core elements of a comprehensive PV curriculum to guide the PV integration into the healthcare schools' curriculum [37]. Besides, Stakeholders' initiated on behalf of the WHO an agreement about PV competencies and key aspects of subjects that should be taught with a focus on clinical Aspects [4]. The five main aspects that were identified include 1) understanding the importance of PV; 2) preventing; 3) recognizing; 4) managing and; 5) reporting ADRs. These competencies should be integrated

into the curriculum of healthcare students to improve their knowledge about PV. This review has helped us gather evidence about the absence of standardized validated measure to evaluate the KAP of PV and ADR reporting of any healthcare school students. Our search strategy was comprehensive, including studies published in English, and a manual search of relevant studies. Besides, quality assessment was conducted to evaluate the quality of the design of the included studies and the presence of potential bias. The main limitation in our review is the heterogeneity of assessment measures used between different included studies, which made a meta-analysis impossible. However, this heterogeneity directs the future need for a standardized validated assessment measure and a unified PV educational intervention for healthcare students in different healthcare schools. Besides, many studies have no information about the missing data within completed questionnaires and have not conducted pilot testing. Another limitation was that the searches developed and carried out without collaboration with a trained information specialist.

## Conclusions

This review demonstrated the lack of PV knowledge among healthcare students and identified several research gaps that need to be focused on future research. These include; developing a standard validated measure to assess students' knowledge, attitude, and perception toward PV. Further, the development of a unified PV education intervention to adequately prepare our future healthcare providers to rationally report ADR of drugs is crucial.

## Supporting information

**S1 Appendix. Search strategy used in MEDLINE.**
(DOCX)

**S2 Appendix. PRISMA 2009 checklist.**
(DOCX)

**S3 Appendix. Questions used to measures knowledge, attitude, and perceptions of medical students in the included studies.**
(DOCX)

## Author Contributions

**Conceptualization:** Monira Alwhaibi, Noha A. Al Aloola.

**Data curation:** Monira Alwhaibi, Noha A. Al Aloola.

**Formal analysis:** Monira Alwhaibi, Noha A. Al Aloola.

**Methodology:** Monira Alwhaibi, Noha A. Al Aloola.

**Writing – original draft:** Monira Alwhaibi, Noha A. Al Aloola.

**Writing – review & editing:** Monira Alwhaibi, Noha A. Al Aloola.

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
