## [Decision Letter · Decision Letter 0]

10 Feb 2020

PONE-D-20-00501

Medical Students' Knowledge, Attitude and Perception of Pharmacovigilance: A Systematic Review

PLOS ONE

Dear Dr. Alwhaibi,

Thank you for submitting your manuscript to PLOS ONE. After careful consideration, we feel that it has merit but does not fully meet PLOS ONE’s publication criteria as it currently stands. Therefore, we invite you to submit a revised version of the manuscript that addresses the points raised during the review process.

Please pay particular attention to the reviewer comments surrounding the PICO and the search strategies, and whether searches were sufficiently comprehensive to identify relevant studies.

We would appreciate receiving your revised manuscript by Mar 26 2020 11:59PM. To enhance the reproducibility of your results, we recommend that if applicable you deposit your laboratory protocols in protocols.io, where a protocol can be assigned its own identifier (DOI) such that it can be cited independently in the future. For instructions see: http://journals.plos.org/plosone/s/submission-guidelines#loc-laboratory-protocols

We look forward to receiving your revised manuscript.

Kind regards,

Lisa Susan Wieland

Academic Editor

PLOS ONE

Journal Requirements:

2. Please ensure that the table describing studies is included in the main body of the manuscript. Ensure the PRISMA checklist is included in the body of the manuscript as Figure 1.

"This research project was supported by a grant from the “Research Center of the Center for Female Scientific and Medical Colleges”, Deanship of Scientific Research, King Saud University." and "The project was fully supported financially by the Vice Deanship of Research Chairs, King Saud University Riyadh, Saudi Arabia."

'The authors received no specific funding for this work'

Reviewers' comments:

Reviewer's Responses to Questions

**Comments to the Author**

1. Is the manuscript technically sound, and do the data support the conclusions?

Reviewer #1: Partly

Reviewer #2: Yes

Reviewer #3: No

2. Has the statistical analysis been performed appropriately and rigorously? 

Reviewer #1: I Don't Know

Reviewer #2: I Don't Know

Reviewer #3: N/A

3. Have the authors made all data underlying the findings in their manuscript fully available?

Reviewer #1: Yes

Reviewer #2: No

Reviewer #3: Yes

4. Is the manuscript presented in an intelligible fashion and written in standard English?

Reviewer #1: Yes

Reviewer #2: Yes

Reviewer #3: Yes

5. Review Comments to the Author

Reviewer #1: 1. Is the manuscript technically sound, and do the data support the conclusions?

The manuscript is technically sound, however this manuscripts aim is not consistent with the outcome of the search and the results. The authors claim to have only included MEDICAL students, however most included articles are NOT about medical students (ie pharmacy, dental or nursing). The conclusion, where you claim knowledge and perceptions about PV in MEDICAL students is insufficient, should therefore be changed.

2. Has the statistical analysis been performed appropriately and rigorously?

The manuscript doesn’t include a section on statistical analysis, although some descriptive statistical tests were probably done.

3. Have the authors made all data underlying the findings in their manuscript fully available?

Yes.

4. Is the manuscript presented in an intelligible fashion and written in standard English?

Yes, most of the manuscript is written in standard English, although many parts should be looked over because of spelling errors or non-fluent sentences (my comments are presented down below).

More specific remarks

Introduction

Page 5, line 8: “Have the above questions adequately… “:This sentence is too complicated and doesn’t flow, please rewrite.

Methods:

Overall: No section was dedicated to the (descriptive) statistical analysis.

Page 5, line 19/20: “Articles search of the four databases was conducted independently by two review authors (MA and NA), the disagreement was resolved by consensus.”: We suggest changing “the disagreement” to “any disagreement”.

Page 5, line 21: “Limits that were applied included selecting studies those published in English from inception until December 2019 …… “: Incorrect grammar. Possible to delete “those”.

Page 6, line 1: “studies that pertaining to the evaluation …..”: Incorrect grammar. Either: use “studies pertaining to” OR “studies that pertained to the ……”

Page 6, line 2/3: “and where the participants are medical schools students but not healthcare professionals”: Too much information, medical students are not healthcare professionals, so choose either “medical school students’ or “not healthcare professionals”.

Page 6, line 13: “This tool covers twenty-two criteria for Study design quality and biases in the study.“ : No need for a capital letter.

Results:

General remarks: Your aim is to analyze MEDICAL students knowledge and perceptions, you have only included the term MEDICAL STUDENT in your search, however have found studies who only include PHARMACY or DENTAL students. Please elaborate on this strange finding. Do you also think PHARMACY and DENTAL students are MEDICAL students. In this case you should clearly state this.

Page 7, line 6/12: Please add more sourceremarks to these lines. E.g. “Twenty studies examined the Pharmacovigilance knowledge and nine evaluated attitudes towards pharmacovigilance and thirteen studies measured the perception of pharmacovigilance”: has no sources.

Page 7, line 19: Wrong word: “Key information form selected articles were extracted and …”, you probably mean “Key information from selected articles were extracted and …”,

Page 8, line 1/3: “The most common questions asked about PV, ADR definitions and the local regulatory body of ADR reporting, while few studies asked about the ADR causality assessment, types of ADR, and the online WHO PV database (Appendix 2).” Incorrect grammar, is missing a verb and is too long.

Page 8, line 11/12: “Attitude towards pharmacovigilance and ADR reporting was measured using multiple-choice response options range from 2 to 5 questions”, should be “ranging”.

Page 8, line 15/16: and 24/1 “The attitude towards PV and ADRs from the included studies was ranging from 25% to 97%”: should be “…. studies ranged from ……”

Discussion

Page 9, line 11/13: ´Moreover, some studies piloted their instrument and some were face and content validated, around 50% measured the reliability, i.e., internal consistency using Cronbach’s Alpha.”:Incorrect grammar and is too long, try making it into two sentences.

Page 9, line 16/19: “This variation could be attributed to the following factors: i) inconsistency in tools/measured used to evaluate KAP between the studies; ii) lacking validated KAP measures, iii) different study setting, and iv) different experience the students have during their education about PV and ADRs reporting.”: point iv) Incorrect grammar. Try “and iv) different PV and ADR reporting experiences during their education.”

Page 9, line 21/22: You mention that the previous published review investigates all healthcare students (these include medical, pharmacy, dental and nursing). This review also covers these “healthcare students” and mostly focusses on pharmacy students in the text.

Page 10, line 2/5: “In addition, many educational interventions have been implemented among medical schools students to improve PV knowledge to enhance perception and positive attitude toward PV such as; short lecture, multiple training workshops, clinical experience in ADR reporting and assessment [3].”: Last example has incorrect grammar.

Page 10, line 5/7: “These interventions improved students PV knowledge and competences to a different extent, however, due to variation in questions or grouped outcomes scores been used the authors were not able to conclude which intervention was the best.”: Incorrect grammar.

Page 10, line 8/10: “In our review, students satisfaction toward PV coverage in their curriculum was vary from 21 to 85% , which indicates variation in PV integration between different medical schools, highlight the need for uniform PV educational intervention.”: it should be: “student satisfaction towards PV”, it should be: “in their curriculum varied from ….”. Also please make two sentences! Please rewrite the second part of the scentence, it doesn’t flow.

Page 10, line 11: There is a WHO-ISoP core curriculum for pharmacovigilance! Jurgen Backmann et all. Teaching Pharmacovigilance: the WHO-ISoP Core Elements of a Comprehensive Modular Curriculum 2014, drug safety.

Page 10, line 11/14: Too long and malformed sentence. Please rewrite.

Page 10, line 18/19: “This review has helped us gather evidence about the lacking of standardized validated measure to evaluate the KAP of PV and ADR reporting of any medical school students.”: Stange word option ( the lacking), maybe change it for “the absence”

Page 10, line 22/23: “The main limitation in our review is the heterogeneity of assessment measures used between different included studies, which band us from conducting a Meta-analysis.”: change word option (band us), maybe use “made a meta-analysis impossible”.

Conclusion

You argue that this review demonstrated the lack of PV knowledge among MEDICAL students, however you have only 5-6 studies on medical students however include a larger amount of studies with dental, pharmacy and nursing students.

Table 1

- It is surprising that you haven’t included any articles form 2019. I already know of a study by Katyal et al 2019, which should fit your search.

- Please elaborate on the difference between a cross-sectional study and a pilot study. I thought you only included cross-sectional studies? If not so, you missed a lot of interventional studies who also published cross-sectional results (ie pre- or post-interventional results).

Reviewer #2: Dear authors: I'm going to stick to the search strategy:

1. The databases searched are appropriate for a systematic review on medical education (MEDLINE & ERIC especially). I typically expect Embase to be included as well for the international literature. Is there a reason this was not included?

2. MedLine should be MEDLINE (in abstract, design & search strategy, and anywhere else mentioned)

3. Please list the platform used for the search of ERIC.

4. I have some concerns about the low numbers of results (~330). For instance, here is how I interpret the PubMed search based on the search description in the methods,

(knowledge OR attitude OR perception) AND medical students AND (pharmacovigilance OR adverse drug reactions reporting)

There are only ~30 results with that search in the version of PubMed that would've been used for this project. While it may have contained every relevant article, it seems far too narrow, especially when I run a more typical search that gets 3x as many:

(medical students[mh] or student*[tiab]) and (pharmacovigilance[mh] or pharmacovigilance[tiab] or pharmaco-vigilance[tiab] or Adverse Drug Reaction Reporting Systems[mh])

I don't mean to say that this is a perfect search, but that it brings in more suggests to me that there may be some missing in your analysis.

5. It's expected for systematic review searches to explicitly include controlled terms (MeSH in the case of PubMed) and more text word variations. For example, plurals (attitudes), other forms (perceive), other words (beliefs, residents, fellows). These are not called out in the methods at all.

6. I applaud the authors for their manual search.

I fear that the search undertaken wasn't comprehensive enough to support the review.

Reviewer #3: There are several issues with the methods of with this review

* Reviewers write that they the review was conducted according to PRISMA, However, PRISMA is about reporting not conducting. For example PRISMA says you must describe all resources searched, but Cochrane Collaboration MECIR lists which databases must be searched.

* The eligibility criteria of included studies was not clearly described. "studies pertaining to the evaluation of...". What types of studies? Were intervention studies going to be included?

*The description of the search is inadequate, and if this was the exact search it is not comprehensive enough to have retrieved all possible articles. Medline, CINAHL, and ERIC all have elaborate thesaurus terms which were not used in the searches. If the databases were searched at the same time, this is also not appropriate as this does not allow the use of thesaurus terms.

*The description of study selection does not provide information required by PRISMA. How many screeners? Was the screening done independently.

*Quality assessment used a reporting standard (STROBE) not an assessment of risk of bias in the methods. Giving the percentage of points received by each article according to strobe does not give me any information about the potential risk of bias across these studies as a whole.

*Figure 1, PRISMA flowchart, belongs in the results section, not the methods section

6. PLOS authors have the option to publish the peer review history of their article (what does this mean?). If published, this will include your full peer review and any attached files.

Reviewer #1: No

Reviewer #2: No

Reviewer #3: No

---

## [Author Response · Author response to Decision Letter 0]

21 Mar 2020

We would like to thank the editor and the reviewers for the time and effort they spent on reviewing our manuscript entitled “Healthcare Students' Knowledge, Attitude and Perception of Pharmacovigilance: A Systematic Review", their valuable and insightful comments have improved our manuscript substantially.

We are excited to have been given the opportunity to revise our manuscript and respond to the revisions. We have gone through all comments received and appropriate changes/amendments have been made correspondingly to the paper (Highlighted) are summarized in the following:

Reviewer(s)' Comments to Author

Reviewer# 1

Comment # 1: The manuscript is technically sound, however this manuscripts aim is not consistent with the outcome of the search and the results. The authors claim to have only included MEDICAL students, however most included articles are NOT about medical students (ie pharmacy, dental or nursing). The conclusion, where you claim knowledge and perceptions about PV in MEDICAL students is insufficient, should therefore be changed.

Response: We thank the reviewer for his valuable comment. The authors meant “Healthcare students (i.e., medical, pharmacy, dental or nursing)” not “Medical students. We now have changed from “Medical students to “Healthcare students” throughout the whole manuscript. 

Comment # 2: The manuscript doesn’t include a section on statistical analysis, although some descriptive statistical tests were probably done.

Response: Thank you for pointing this out. Now we have included the statistical analysis part in (Page 7, line 8-10)

Comment # 3: Introduction: Page 5, line 8: “Have the above questions adequately… “:This sentence is too complicated and doesn’t flow, please rewrite.

Response: We have rewritten the sentence to make it clear to the reader (Page 5, line 8-9). 

Comment # 4: Methods: Overall: No section was dedicated to the (descriptive) statistical analysis.

Page 5, line 19/20: “Articles search of the four databases was conducted independently by two review authors (MA and NA), the disagreement was resolved by consensus.”: We suggest changing “the disagreement” to “any disagreement”.

Response: Now we have included the statistical analysis part in (Page 7, line 8-10). We have also changed “the disagreement” to “any disagreement” (Page 6, line 11).

Comment # 5: Methods:

Page 5, line 21: “Limits that were applied included selecting studies those published in English from inception until December 2019 …… “: Incorrect grammar. Possible to delete “those”.

Page 6, line 1: “studies that pertaining to the evaluation …..”: Incorrect grammar. Either: use “studies pertaining to” OR “studies that pertained to the ……”

Page 6, line 13: “This tool covers twenty-two criteria for Study design quality and biases in the study.“ : No need for a capital letter.

Response: Thanks for the correction (Page 6, line 12, 13) (Page 7, line 6). 

Comment # 6: Methods: Page 6, line 2/3: “and where the participants are medical schools students but not healthcare professionals”: Too much information, medical students are not healthcare professionals, so choose either “medical school students’ or “not healthcare professionals”.

Response: We made the suggested change (Page 6, 14). 

Comment # 7: Results: General remarks: Your aim is to analyze MEDICAL students knowledge and perceptions, you have only included the term MEDICAL STUDENT in your search, however have found studies who only include PHARMACY or DENTAL students. Please elaborate on this strange finding. Do you also think PHARMACY and DENTAL students are MEDICAL students. In this case you should clearly state this.

Response: We meant healthcare students which include MEDICAL, PHARMACY, DENTAL, and NURSING. Therefore, now we have changed it from “Medical students” to “Healthcare students” throughout the whole manuscript. To make it clear to the reader we have included the methods section the eligibility criteria (Page 5, line 12-17).

Comment # 8: Results: Page 7, line 6/12: Please add more source remarks to these lines. E.g. “Twenty studies examined the Pharmacovigilance knowledge and nine evaluated attitudes towards pharmacovigilance and thirteen studies measured the perception of pharmacovigilance”: has no sources.

Response: Change has been made (Page 8, line 3-6). 

Comment # 9: Results:

Page 8, line 11/12: “Attitude towards pharmacovigilance and ADR reporting was measured using multiple-choice response options range from 2 to 5 questions”, should be “ranging”.

10: Page 8, line 15/16: and 24/1 “The attitude towards PV and ADRs from the included studies was ranging from 25% to 97%”: should be “…. studies ranged from ……”

Response: Changes has been made (Page 9, line 10, 14)

Comment # 10: Discussion: Page 9, line 11/13: ´Moreover, some studies piloted their instrument and some were face and content validated, around 50% measured the reliability, i.e., internal consistency using Cronbach’s Alpha.”: Incorrect grammar and is too long, try making it into two sentences.

Response: We made the suggested change (Page 10, line 10-11). 

Comment # 11: Discussion: Page 9, line 16/19: “This variation could be attributed to the following factors: i) inconsistency in tools/measured used to evaluate KAP between the studies; ii) lacking validated KAP measures, iii) different study setting, and iv) different experience the students have during their education about PV and ADRs reporting.”: point iv) Incorrect grammar. Try “and iv) different PV and ADR reporting experiences during their education.”

Response: Thanks for the correction (Page 10, line 16-17). 

Comment # 12: Discussion: Page 9, line 21/22: You mention that the previous published review investigates all healthcare students (these include medical, pharmacy, dental and nursing). This review also covers these “healthcare students” and mostly focusses on pharmacy students in the text.

Response: The previously published review hase included cross sectional and intervention studies to identify effective educational interventions that promote pharmacovigilance early in their education and career among pharmacy and medical students. However, our review was to evaluate the literature that measures the level of healthcare students’ knowledge, attitude and perception of pharmacovigilance and ADRs reporting. 

The focus of this review was on healthcare students in general, however, most of the retrieved published studies were among pharmacy students.

Comment # 13: Discussion: 

Page 10, line 2/5: “In addition, many educational interventions have been implemented among medical schools students to improve PV knowledge to enhance perception and positive attitude toward PV such as; short lecture, multiple training workshops, clinical experience in ADR reporting and assessment [3].”: Last example has incorrect grammar.

Page 10, line 5/7: “These interventions improved students PV knowledge and competences to a different extent, however, due to variation in questions or grouped outcomes scores been used the authors were not able to conclude which intervention was the best.”: Incorrect grammar.

Response: These sentences have been corrected (Page 11, line 1-4) 

Comment # 14: Discussion: Page 10, line 8/10: “In our review, students satisfaction toward PV coverage in their curriculum was vary from 21 to 85% , which indicates variation in PV integration between different medical schools, highlight the need for uniform PV educational intervention.”: it should be: “student satisfaction towards PV”, it should be: “in their curriculum varied from ….”. Also please make two sentences! Please rewrite the second part of the scentence, it doesn’t flow.

Response: Changes have been made (Page 11, line 5-7). 

Comment # 15: Discussion: Page 10, line 11: There is a WHO-ISoP core curriculum for pharmacovigilance! Jurgen Backmann et all. Teaching Pharmacovigilance: the WHO-ISoP Core Elements of a Comprehensive Modular Curriculum 2014, drug safety.

Response: We thank the reviewer for suggesting adding this reference (Page 11, line 9-12).

Comment # 16: Discussion: Page 10, line 11/14: Too long and malformed sentence. Please rewrite.

Response: We have rewritten to shorten and improve the sentence (Page 11, line 13-14). 

Comment # 17: Discussion:

Page 10, line 18/19: “This review has helped us gather evidence about the lacking of standardized validated measure to evaluate the KAP of PV and ADR reporting of any medical school students.”: Stange word option ( the lacking), maybe change it for “the absence”

Page 10, line 22/23: “The main limitation in our review is the heterogeneity of assessment measures used between different included studies, which band us from conducting a Meta-analysis.”: change word option (band us), maybe use “made a meta-analysis impossible”.

Response: Changes have been made (Page 11, line 18, 23). 

Comment # 18: Conclusion

You argue that this review demonstrated the lack of PV knowledge among MEDICAL students, however you have only 5-6 studies on medical students however include a larger amount of studies with dental, pharmacy and nursing students.

Response: We meant the “healthcare students”, now we have corrected the wording throughout the whole manuscript.

Comment # 19: Table 1

- It is surprising that you haven’t included any articles form 2019. I already know of a study by Katyal et al 2019, which should fit your search.

- Please elaborate on the difference between a cross-sectional study and a pilot study. I thought you only included cross-sectional studies? If not so, you missed a lot of interventional studies who also published cross-sectional results (ie pre- or post-interventional results).

Response: As per the second reviewer’s suggestion, we have searched the EMBASE and included some recent articles in table 1. We have excluded interventional studies as the focus of this review was to evaluate the students’ knowledge, attitude, and perception of PV. 

Reviewer# 2

Dear authors: I'm going to stick to the search strategy

Comment # 1: The databases searched are appropriate for a systematic review on medical education (MEDLINE & ERIC especially). I typically expect Embase to be included as well for the international literature. Is there a reason this was not included?

Response: We totally agree with the reviewer, initially the Embase database was not available by our institute. However, now we have requested to access this database and updated the search results (Page 6, line 1)(Figure 1). 

Comment # 2: MedLine should be MEDLINE (in abstract, design & search strategy, and anywhere else mentioned)

Response: Now we have made the correction (Page 2, line 5), (Page 5, line 21).

Comment # 3: Please list the platform used for the search of ERIC.

Response: The platform used for the search of ERIC was EBSCO (Page 6, line 3).

Comment # 4: I have some concerns about the low numbers of results (~330). For instance, here is how I interpret the PubMed search based on the search description in the methods,

(knowledge OR attitude OR perception) AND medical students AND (pharmacovigilance OR adverse drug reactions reporting)

There are only ~30 results with that search in the version of PubMed that would've been used for this project. While it may have contained every relevant article, it seems far too narrow, especially when I run a more typical search that gets 3x as many:

(medical students[mh] or student*[tiab]) and (pharmacovigilance[mh] or pharmacovigilance[tiab] or pharmaco-vigilance[tiab] or Adverse Drug Reaction Reporting Systems[mh])

I don't mean to say that this is a perfect search, but that it brings in more suggests to me that there may be some missing in your analysis.

Response: Thank you for pointing this out. As per the reviewer’s suggestion, we have redone the systematic search, updated the search strategy, results, figure, and tables. Appendix 1 shows the complete search strategy used in Medline.

Comment # 5: It's expected for systematic review searches to explicitly include controlled terms (MeSH in the case of PubMed) and more text word variations. For example, plurals (attitudes), other forms (perceive), other words (beliefs, residents, fellows). These are not called out in the methods at all.

Response: These search terms have been used but have not been described adequately in the methods section. Now, more description is added “Research articles were retrieved from five databases (MEDLINE and CINAHL via EBSCO, EMBASE, ERIC, and Cochrane Database of Systematic Reviews via OVID) with database-specific queries. These databases were searched using both controlled and free-text language. In terms of free-text search, the keywords included the following terms/combination of terms: (knowledge OR attitude OR perception) AND (healthcare students OR medical students OR pharmacy students OR dental) AND (pharmacovigilance OR adverse drug reactions reporting). For the controlled language search included the following exploded MeSH terms: "knowledge”, “attitude”, “perception", "Students, Medical", "Students, Pharmacy", "Students, Nursing", "Students, Dental", "Pharmacovigilance" and " Drug-Related Side Effects and Adverse Reactions” as recommended for each databases (Page 6, line 2-9). Other terms have also been used “such as beliefs or views or feelings or experience”

Comment # 6: I applaud the authors for their manual search.

Response: Thank you.

Comment # 7: I fear that the search undertaken wasn't comprehensive enough to support the review.

Response: We have redone the systematic search taking into consideration the reviewer’s suggestions above (EMBASE). Corresponding changes have been made in the manuscript as well as the figure and tables.

Reviewer# 3

There are several issues with the methods of with this review

Comment # 1: Reviewers write that they the review was conducted according to PRISMA, However, PRISMA is about reporting not conducting. For example PRISMA says you must describe all resources searched, but Cochrane Collaboration MECIR lists which databases must be searched.

Response: We thank the reviewer for pointing this out. Now we have changed the wording to “The present systematic review was reported according to PRISMA” (Page 5, line 19-21).

Comment # 2: The eligibility criteria of included studies was not clearly described. "studies pertaining to the evaluation of...". What types of studies? Were intervention studies going to be included?

Response: Now we have included the eligibility criteria (Page 5, line 12-17).

Comment # 3: The description of the search is inadequate, and if this was the exact search it is not comprehensive enough to have retrieved all possible articles. Medline, CINAHL, and ERIC all have elaborate thesaurus terms which were not used in the searches. If the databases were searched at the same time, this is also not appropriate as this does not allow the use of thesaurus terms.

Response: We thank the reviewer for this comment. Each database was searched individually with database-specific queries. We agree with the reviewer that the search terms used have not been described adequately in the methods section. Now, more description is added “Research articles were retrieved from five databases (MEDLINE and CINAHL via EBSCO, EMBASE, ERIC, and Cochrane Database of Systematic Reviews via OVID) with database-specific queries. These databases were searched using both controlled and free-text language. In terms of free-text search, the keywords included the following terms/combination of terms: (knowledge OR attitude OR perception) AND (healthcare students OR medical students OR pharmacy students OR dental) AND (pharmacovigilance OR adverse drug reactions reporting). For the controlled language search included the following exploded MeSH terms: "knowledge”, “attitude”, “perception", "Students, Medical", "Students, Pharmacy", "Students, Nursing", "Students, Dental", "Pharmacovigilance" and " Drug-Related Side Effects and Adverse Reactions” as recommended for each databases” (Page 6, line 2-11). Appendix 1 shows the complete search strategy used in Medline.

Comment # 4: The description of study selection does not provide information required by PRISMA. How many screeners? Was the screening done independently?

Response: Literature screening has been done independently. Now, we clarified it in the methods section (Page 6, line 16-18).

Comment # 5: Quality assessment used a reporting standard (STROBE) not an assessment of risk of bias in the methods. Giving the percentage of points received by each article according to strobe does not give me any information about the potential risk of bias across these studies as a whole.

Response: STROBE is one of the assessment tools used in observational studies.1 It has item £ 9 that measures the risk of bias in each included study. However, it does not give any information about the potential risk of bias across these studies as a whole.

1. Sanderson, S., Tatt, I.D. and Higgins, J., 2007. Tools for assessing quality and susceptibility to bias in observational studies in epidemiology: a systematic review and annotated bibliography. International journal of epidemiology, 36(3), pp.666-676.

Comment # 6: Figure 1, PRISMA flowchart, belongs in the results section, not the methods section.

Response: Thank you for this suggestion (Page 6, line 21)

---

## [Editor Report · Decision Letter 1]

8 Apr 2020

PONE-D-20-00501R1

Healthcare Students' Knowledge, Attitude and Perception of Pharmacovigilance: A Systematic Review

PLOS ONE

Dear Dr. Alwhaibi,

Thank you for submitting your manuscript to PLOS ONE. After careful consideration, we feel that it has merit but does not fully meet PLOS ONE’s publication criteria as it currently stands. Therefore, we invite you to submit a revised version of the manuscript that addresses the points raised during the review process.

We would appreciate receiving your revised manuscript by May 23 2020 11:59PM. To enhance the reproducibility of your results, we recommend that if applicable you deposit your laboratory protocols in protocols.io, where a protocol can be assigned its own identifier (DOI) such that it can be cited independently in the future. For instructions see: http://journals.plos.org/plosone/s/submission-guidelines#loc-laboratory-protocols

We look forward to receiving your revised manuscript.

Kind regards,

Lisa Susan Wieland

Academic Editor

PLOS ONE

Additional Editor Comments (if provided):

The manuscript is improved, however some of the reviewer comments have not yet been adequately addressed. See comments here:

1) The search strategy is improved. However, if you did not have the searches developed and carried out in collaboration with a trained information specialist please add this to the limitations of the study, as you cannot be certain that there were not additional studies that you did not retrieve. Also, did you search the Cochrane Database of Systematic Reviews or was it the Cochrane Library? If you were searching for systematic reviews it was probably the Cochrane Database of Systematic Reviews but if you were searching for individual controlled trials it was probably the Cochrane Central Register of Controlled Trials. Both of these databases are in the Cochrane Library.

2) The rationale for including a ‘pilot study’ in which pre and post tests were administered and excluding studies reporting KAP measures pre- and post- some intervention is unclear. What is the difference between assessing the results of an intervention and testing the effects of usual education? This needs to be clarified. The inclusion of cross-sectional studies is clear, but the cohort studies could mean anything from a single group tested at multiple time points to a randomized controlled trial.

3) The reviewer is correct that STROBE is a reporting guideline. The authors of the article you cite say that STROBE was referred to in many of the assessment instruments because many STROBE items may have been selected because of presumed association with risk of bias. However the authors do not suggest that STROBE should be used as a risk of bias assessment tool. The authors state that ‘Just under three-quarters of all tools were proposed as being suitable for future use, including all of the critical appraisal tools and generic systematic review tools and six of the tools originally designed for use in a specific systematic review.’ Look in Tables 5-7 for the tools that meet the important criteria identified by the authors (selection of participants, measurement of variables, and control of confounding). These are the tools that the authors judge as appropriate.

A more recent and more targeted approach to risk of bias for surveys may be found in https://www.evidencepartners.com/wp-content/uploads/2017/09/Risk-of-Bias-Instrument-for-Cross-Sectional-Surveys-of-Attitudes-and-Practices.pdf. See https://www.evidencepartners.com/wp-content/uploads/2017/04/Methods-Commentary-Risk-of-Bias-in-cross-sectional-surveys-of-attitude....pdf for comments and further guidance on this tool.

For the pre-post surveys, you might have to look further for a suitable tool, or you could decide to assess the risk of bias based on either the pretest or the posttest survey if you are not interested in the effects of the education intervention. However, as I mentioned above, the inclusion of pretest/posttest pilot studies but the exclusion of other interventional studies lacks a rationale and this needs to be addressed.
---

## [Author Response · Author response to Decision Letter 1]

10 Apr 2020

We would like to thank the editor and the reviewers for the time and effort they spent on reviewing our revised manuscript entitled “Healthcare Students' Knowledge, Attitude and Perception of Pharmacovigilance: A Systematic Review"

All appropriate changes/amendments have been made correspondingly to the paper (Highlighted) are summarized in the following:

Reviewer Comments to Authors

Comment # 1: The search strategy is improved. However, if you did not have the searches developed and carried out in collaboration with a trained information specialist please add this to the limitations of the study, as you cannot be certain that there were not additional studies that you did not retrieve. 

Response: Thank you for the suggestion; we have added this limitation to page 12, lines 1-2

Also, did you search the Cochrane Database of Systematic Reviews or was it the Cochrane Library? If you were searching for systematic reviews it was probably the Cochrane Database of Systematic Reviews but if you were searching for individual controlled trials it was probably the Cochrane Central Register of Controlled Trials. Both of these databases are in the Cochrane Library.

Response: We have searched the Cochrane Database of Systematic Review (Page 6, line 3).

Comment # 2: The rationale for including a ‘pilot study’ in which pre and post tests were administered and excluding studies reporting KAP measures pre- and post- some intervention is unclear. What is the difference between assessing the results of an intervention and testing the effects of usual education? This needs to be clarified. 

Response: The reason for excluding interventional studies was because we are not evaluating the impact of educational material on knowledge, attitude, and perceptions. The reason for including pilot studies, because these studies are usually conducted to evaluate the validity, readability, and reliability of the instruments used in the survey usually among 20-30 participants (Results from the pilot subjects are not included in the study). For example, Limaye et al. study was pilot-tested to a sample of 30 pharmacy students (Pilot subjects (i.e., 30) were not part of the final study that was conducted among 352 students).

The inclusion of cross-sectional studies is clear, but the cohort studies could mean anything from a single group tested at multiple time points to a randomized controlled trial.

Response: We totally agree with the reviewer, therefore we have removed cohort study from the inclusion criteria.

Comment # 3: The reviewer is correct that STROBE is a reporting guideline. The authors of the article you cite say that STROBE was referred to in many of the assessment instruments because many STROBE items may have been selected because of presumed association with risk of bias. However the authors do not suggest that STROBE should be used as a risk of bias assessment tool. The authors state that ‘Just under three-quarters of all tools were proposed as being suitable for future use, including all of the critical appraisal tools and generic systematic review tools and six of the tools originally designed for use in a specific systematic review.’ Look in Tables 5-7 for the tools that meet the important criteria identified by the authors (selection of participants, measurement of variables, and control of confounding). These are the tools that the authors judge as appropriate.

A more recent and more targeted approach to risk of bias for surveys may be found in https://www.evidencepartners.com/wp-content/uploads/2017/09/Risk-of-Bias-Instrument-for-Cross-Sectional-Surveys-of-Attitudes-and-Practices.pdf. 

See https://www.evidencepartners.com/wp-content/uploads/2017/04/Methods-Commentary-Risk-of-Bias-in-cross-sectional-surveys-of-attitude....pdf for comments and further guidance on this tool.

Response: We thank the reviewer for the detailed explanation about this important point and for providing the reference. Now, we have incorporated under the quality assessment the five items risk of bias in cross-sectional surveys of attitudes and practice (Page 7, lines 7-8) (Page 8, lines 15-16).

Reference: Agarwald, A., G. Guyatt, and J. Busse. "Methods commentary: Risk of bias in cross-sectional surveys of attitudes and practices." (2019).

For the pre-post surveys, you might have to look further for a suitable tool, or you could decide to assess the risk of bias based on either the pretest or the posttest survey if you are not interested in the effects of the education intervention. However, as I mentioned above, the inclusion of pretest/posttest pilot studies but the exclusion of other interventional studies lacks a rationale and this needs to be addressed.

Response: Response to this comment was hopefully explained in response to the second comment.

---

## [Editor Report · Decision Letter 2]

5 May 2020

Healthcare Students' Knowledge, Attitude and Perception of Pharmacovigilance: A Systematic Review

PONE-D-20-00501R2

Dear Dr. Alwhaibi,

We are pleased to inform you that your manuscript has been judged scientifically suitable for publication and will be formally accepted for publication once it complies with all outstanding technical requirements.

With kind regards,

Lisa Susan Wieland

Academic Editor

PLOS ONE
---

## [Editor Report · Acceptance letter]

7 May 2020

PONE-D-20-00501R2 

Healthcare Students' Knowledge, Attitude and Perception of Pharmacovigilance: A Systematic Review 

Dear Dr. Alwhaibi:

I am pleased to inform you that your manuscript has been deemed suitable for publication in PLOS ONE. Congratulations! Your manuscript is now with our production department. 

With kind regards,

on behalf of

Dr. Lisa Susan Wieland 

Academic Editor

PLOS ONE